# Attachment-Based Compassion Therapy for Reducing Anxiety and Depression in Fibromyalgia

**DOI:** 10.3390/ijerph19138152

**Published:** 2022-07-02

**Authors:** Alicia Santos, Iris Crespo, Adrián Pérez-Aranda, María Beltrán-Ruiz, Marta Puebla-Guedea, Javier García-Campayo

**Affiliations:** 1Endocrinology and Internal Medicine Departments, Hospital Sant Pau, Universitat Autònoma de Barcelona (UAB), 08025 Barcelona, Spain; asantos@santpau.cat; 2Centro de Investigación Biomédica en Red de Enfermedades Raras (CIBERER, Unit 747), 08025 Barcelona, Spain; icrespo@uic.es; 3Institut d’Investigació Biomèdica-Sant Pau (IIB-SANT PAU), 08025 Barcelona, Spain; 4Department of Basic Sciences, School of Medicine and Health Sciences, Universitat Internacional de Catalunya (UIC), 08195 Barcelona, Spain; 5Institute of Health Research of Aragon (IIS Aragón), Miguel Servet University Hospital, 50009 Zaragoza, Spain; maria.beltran.ruiz@gmail.com (M.B.-R.); martapueblag@gmail.com (M.P.-G.); jgarcamp@gmail.com (J.G.-C.); 6Department of Basic, Developmental and Educational Psychology, Autonomous University of Barcelona, 08193 Cerdanyola del Vallès, Spain; 7Research Network on Chronicity, Primary Care and Health Promotion (RICAPPS) RD21/0016/0005, 50015 Zaragoza, Spain; 8Psychiatry Department, Faculty of Medicine, University of Zaragoza, 50009 Zaragoza, Spain

**Keywords:** attachment-based compassion therapy, fibromyalgia, self-compassion, decentering, anxiety, depression

## Abstract

Fibromyalgia patients often experience anxiety and depressive symptoms; however, validated interventions show only limited efficacy. This pilot study analyzed the effects of a 16-session version of attachment-based compassion therapy (ABCT-16) for improving anxiety and depressive symptomatology, as well as self-compassion and decentering, in 11 fibromyalgia patients. Scales were assessed at four time points: baseline, after sessions 8 and 16, and 3.5 months after the completion of the program. Significant improvements were found in all outcomes after the program, and most remained significant in the follow-up assessment. Our preliminary results suggest that ABCT-16 can be effective for improving anxiety and depressive symptomatology in fibromyalgia patients. Nonetheless, further studies with larger samples and control groups are necessary to confirm these results.

## 1. Introduction

Fibromyalgia is a chronic-pain syndrome characterized by symptoms of widespread musculoskeletal pain, stiffness, sleep disturbances, fatigue, and perceived cognitive difficulties [1,2,3]. It affects approximately 2% of the general population and is often concomitant with other medical disorders, such as gastrointestinal diseases and other pain-related conditions, and mental health disorders [2,3,4]. The specific etiology of fibromyalgia remains unknown, although it has been suggested that psychological, social, and somatic factors may interact and have a role in the predisposition and perpetuation of fibromyalgia symptoms [2].

Different interventions have demonstrated efficacy for reducing fibromyalgia symptomatology, although mostly with small effect sizes and limited long-term effectiveness [5]. Therefore, further research is required into new treatments, among which compassion-based interventions are seen as a promising approach. Compassion refers to an orientation of the mind characterized by sensitivity towards suffering and a commitment to relieve it by recognizing its universality and the ability to meet that pain with equanimity [6]. Higher levels of compassion have been associated with reduced anxiety and depression [7], and different interventions based on promoting compassion and related constructs, such as decentering (i.e., the ability to observe one’s thoughts and feelings in a detached manner) [8], have been developed. Their effectiveness on different outcomes, including anxiety and depression, has been reported [9].

To date, the only compassion-based intervention to be tested on fibromyalgia patients has been attachment-based compassion therapy (ABCT) [10], an intervention that addresses the regulation of attention processes in order to replace self-critical tendencies with self-compassionate attitudes via the development of a more secure attachment figure [11]. ABCT is framed in attachment theory but also includes elements from other compassion-based interventions, such as compassion-focused therapy [12], and incorporates techniques from other psychotherapies, such as mindfulness-based interventions, acceptance and commitment therapy, and dialectical behavior therapy, which have also been reported as helpful techniques for improving depression and anxiety symptoms in patients with fibromyalgia [13,14,15]. The study conducted by Montero-Marín et al. [16] concluded that ABCT improved fibromyalgia symptomatology compared with a relaxation program and that this effect was maintained at a 3-month follow-up, which suggests the important therapeutic potential of ABCT and the need for replication in new studies. Montero-Marín et al. used the original ABCT protocol (i.e., 2 h weekly sessions for 8 weeks), and although they reported a notable adherence to the intervention (i.e., 81.5% of sessions attended, on average) and significant effects, some adaptations of the program could contribute to a better understanding and content acquisition by fibromyalgia patients, owing to characteristics of their disorder that could hinder their commitment to the psychotherapy, i.e., on the one hand, generalized pain and stiffness that makes it hard for them to maintain their posture for long periods of time (e.g., sitting on a chair for 2 h listening to the instructor), and on the other hand, cognitive impairment that entails difficulties in abilities such as attention and memory, referred to as fibrofog, which could hinder their understanding and application of the concepts of the intervention [17].

The aim of the present work was to conduct a preliminary study into the efficacy of an adapted version of ABCT (i.e., ABCT-16) to reduce anxiety and depression in a sample of patients with fibromyalgia, while enhancing self-compassion and decentering. The study hypotheses were (1) that ABCT-16 would significantly reduce posttreatment symptomatology and promote “third wave” skills such as self-compassion and decentering, (2) that these changes would be maintained at the follow-up assessment (i.e., 3.5 months after the end of the program), and (3) that adherence to the intervention would be similar to or higher than that reported by Montero-Marín et al. [16].

## 2. Materials and Methods

### 2.1. Participants

Patients who belonged to a fibromyalgia association in Barcelona, Spain, were contacted and offered the possibility to participate in this pilot study. Of the 80 members of the association, 11 women (mean age of 54.91, SD = 8.81) volunteered and were recruited after checking their compliance with the following inclusion criteria: (1) aged 18 years and over; (2) self-reported diagnoses of fibromyalgia; (3) no prior experience with compassion-based interventions; (4) Spanish language fluency; (5) signed informed consent. The study was conducted during the first six months of 2016. Four assessment points were stablished: baseline, after session 8, posttreatment, and 3.5 months after the completion of the program. The study protocol was approved by the ethical review board of the regional health authority of Aragon, Spain (PI15/0049; 01/04/2015). All procedures performed in this study were in accordance with the 1964 Helsinki Declaration and its later amendments.

### 2.2. Intervention

The ABCT-16 program is an adaptation of the original ABCT protocol developed by García-Campayo and Demarzo [10] that consists of 16 sessions 1 h in duration instead of the original 8 sessions 2 h in duration. This adaptation was introduced after acknowledging the difficulties experienced by a significant proportion of fibromyalgia patients regarding cognitive faculties such as attention and memory; shortening the duration of each session was considered a solution to facilitate understanding of the concepts and reduce dropout rates. Nonetheless, the ABCT-16 program also included homework tasks, and participants received a CD containing a recording of the guided meditation practices of the program during the first session. Guides to home practice were delivered in written form at the end of every session. If a patient was not able to attend a session, a summary of the session, together with a link to the meditations and homework, was sent by email. The program was taught by A.S., a psychologist who had received specific training in ABCT and had experience in mindfulness interventions, with the support of a second psychologist (I.C.). The specific contents of each session and homework tasks can be found in the Appendix A.

### 2.3. Measures

The Hospital Anxiety and Depression Scale (HADS) is a 14-item questionnaire divided in two subscales that assess depression and anxiety levels [18,19]. The total score ranges from 0 to 42, and each subscale (HADS-Anxiety and HADS-Depression) includes 7 items with scores ranging from 0 to 21. Higher scores represent higher levels of depression or anxiety. The Spanish version of the HADS presents strong psychometric properties in the general population and in patients with fibromyalgia, with high internal consistency (α = 0.80–0.85) [20,21].

The Self-Compassion Scale-Short Form (SCS-SF) [7] is a 12-item measure that is a shortened version of the original scale. The answer to each question is recorded on a 5-point Likert scale. The SCS includes three facets of self-compassion that were analyzed in the present study: “self-kindness” (treating oneself with kindness when experiencing suffering), “common humanity” (knowing that suffering is common to all humankind), and “mindfulness” (being aware of suffering, in oneself and others). The scores of SCS and its subscales range from 1 to 5, where higher scores represent higher levels of self-compassion. The Spanish version of the SCS-SF shows good internal consistency (α = 0.86) [22].

The Experiences Questionnaire (EQ-D) is an 11-item measure to assess decentering, defined as “the ability to observe one’s thoughts and feelings in a detached manner” [8,23]. The answer to each question is recorded on a 5-point Likert scale. Total scores range from 11 to 55, where higher scores represent higher levels of decentering. The psychometric properties of the Spanish version of the EQ-D are reported to be adequate [8].

### 2.4. Data Analysis

The effects of ABCT were explored using the Wilcoxson test, comparing each timepoint with the baseline score. Rosenthal’s r equivalent was used to calculate effect sizes. Moreover, the reliable change index (RCI) and the clinically significant change (CSC) criterium were calculated for the HADS [24]. Normative data from nonpatients were used to establish whether patients had experienced improvements of their anxiety and depression symptoms [25]. The RCI was calculated according to the following formula:RCI = (Xpre − Xpost)/(S diff), where Sdiff = √(2 ∗ (SE))2, and SE = SD ∗ √(1 − r)
where Xpre = pretreatment group mean; Xpost = posttreatment group mean; SE = standard error; SD = standard deviation; r = reliability of the measurement (Cronbach’s α).

Baselines for postintervention difference of ≥2.80, 4.02, and 5.11 points were considered reliable improvements of the HADS “Anxiety,” HADS “Depression,” and HADS total scores, respectively. Additionally, patients were considered recovered if their posttreatment scores were below the CSC criterium: CSC = ((SD normative data)(Xpatients) + (SDpatients)(Xnormative data))/(SDnormative data + SDpatients)

The cutoff points were 14.67 for HADS “Anxiety,” 19.43 for HADS “Depression,” and 34 for HADS total scores. The SPSS 22 software package (IBM Corp., Armonk, NY, USA) was used for the statistical analyses. Statistical differences were considered significant when *p* < 0.05.

## 3. Results

### 3.1. Adherence to the ABCT Program

The 11 participants who commenced the intervention were evaluated at every assessment point. The average number of sessions attended was 10.4 (*SD* = 2.2), and all the participants completed at least half of the program (i.e., attended 8 sessions). On average, the participants completed 64.8% of the sessions.

### 3.2. Efficacy of ABCT on Anxiety and Depression

Raw scores are presented in Table 1. The Wilcoxson test indicated that ABCT produced significant reductions in the HADS and its subscales at all the assessment points. The effect sizes increased from session 8 to posttreatment and were maintained in the follow-up (see Table 2).

Posttreatment, eight patients (72.7%) reduced their HADS “Anxiety” baseline score by 2.80 points or more; five (45.5%) reduced their HADS “Depression” baseline scores by 4.08 points or more; and eight (72.7%) reduced their total score by 5.11 points or more. With regard to CSC, only two patients presented baseline levels of anxiety higher than the cutoff point, and both reduced them posttreatment; no patients had baseline levels of depression higher than the cutoff point or higher than overall symptomatology.

### 3.3. Efficacy of ABCT on Self-Compassion and Decentering

The Wilcoxson test indicated that ABCT produced significant improvements in SCS-SF posttreatment, but not after 8 sessions, nor at the follow-up assessment. The “self-kindness” subscale experienced a significant increase posttreatment, and this was maintained at follow-up; the “common humanity” subscale did not show any significant effect, and “mindfulness” experienced a significant improvement after session 8, which was enhanced posttreatment and maintained after 3.5 months. With regard to the EQ-D, significant improvements were observed after 8 sessions that were enhanced posttreatment and maintained at follow-up. These results are detailed in Table 2.

## 4. Discussion

This pilot study aimed to explore the efficacy of an adapted version of ABCT for improving anxiety and depression symptoms in a sample of fibromyalgia patients and to study its effects on self-compassion and decentering. 

Our preliminary results suggest that ABCT-16 reduced both anxiety and depressive symptomatology after 8 sessions, i.e., after half of the program. The effect sizes, however, increased once the intervention was completed, and the follow-up assessment indicated that they were maintained after 3.5 months. A very high proportion of patients achieved a reliable improvement (72.7% for anxiety and 45.5% for depression). However, CSC was more difficult to assess, considering that baseline symptomatology levels were relatively low; only two patients presented high enough HADS “Anxiety” baseline scores, and although they both experienced posttreatment CSC, further research using larger samples and with more severe symptomatology is needed to assess with greater accuracy the potential of ABCT-16 to produce clinically significant changes in both anxiety and depressive symptomatology.

Montero-Marín et al. [16] found that the original ABCT program (i.e., 8 sessions 2 h in duration) was superior to an active control group (i.e., relaxation program) in the main clinical outcomes, including functional status, as measured by the Fibromyalgia Impact Questionnaire [26], and anxiety and depressive symptoms, as measured by the HADS. These improvements were maintained at the 3-month follow-up assessment, which was a particularly important finding and was corroborated in our study; ABCT seems to produce lasting effects on different outcomes, which could be seen as a very valuable aspect, considering the typical loss of effect commonly found with many psychotherapies, including ”third wave” interventions [14,27]. The reasoning behind the long-term impact of ABCT could be that this intervention focuses on producing a deep change in the way individuals relate to themselves by promoting the development of a more secure attachment figure. While this therapeutic change might be harder to produce than others that are more context related, its effects could last longer once it has been achieved [10].

On the other hand, patients also presented significant posttreatment improvements in terms of the self-compassion facets of “self-kindness” and “mindfulness,” which were maintained at the follow-up assessment. A previous study [28] already reported significant effects of ABCT on self-compassion and mindfulness facets. Moreover, these “third wave” variables turned out to be significant mediators of the change brought about by the intervention on different outcomes. Self-compassion played an intermediary role in the effect of ABCT on patients’ functional status; “common humanity,” defined as the acknowledgement of the fact that all human beings are united by the experience of suffering, was a significant mediator of the changes produced in anxiety symptoms; and “mindfulness,” as a concept opposite to “over-identification,” was a significant mediator of the improvements observed in depressive symptoms.

The latter result connects with another outcome of our study, i.e., decentering is strongly related to mindfulness in the context of self-compassion [29], as it refers to the capacity of feeling painful emotions and thoughts without over-identifying with them. Improvements of decentering indicated that patients at the end of the program were able to observe their thoughts and feelings from a more detached perspective, not necessarily reflecting reality. Interestingly, decentering scores significantly correlate with anxiety and depression scores in patients with chronic pain, suggesting that the ability to decenter plays a role in the perpetuation of anxiety and depression symptoms [30].

The ABCT-16 program was an adaptation of an original protocol designed to meet the needs of fibromyalgia patients, who often present with cognitive difficulties (i.e., fibrofog) that hinder their ability to focus on the instructors’ explanations and who acknowledged in previous studies that the length of the psychotherapy sessions––around 2 h––was an obstacle because it was hard for many of them to remain seated for such a long period [26]. The rates of adherence to ABCT-16 in this pilot study were good, given that all patients attended at least 8 of the 16 sessions. The role of the instructors was key in maintaining the patients’ link with the intervention, since they conducted phone calls and sent the session contents and homework via email every time a patient was not able to attend. Thus, it remains unclear whether adherence to the intervention would have been equally high had the instructors not made such efforts. Future studies using larger samples and control groups could put this to the test. Moreover, given that previous studies have already found the effects of ABCT to be maintained after 3 months [16], further studies should consider extending the follow-up assessments to at least 12 months in order to verify whether the hypothesized deep changes produced in the attachment figure of the individual remain significant in the longer term.

Our preliminary findings need to be interpreted in light of the study limitations. First, the sample size was very small; therefore, it was not representative of the population of fibromyalgia patients, which hindered more sophisticated analyses and the generalization of our findings. Fibromyalgia diagnoses were not assessed by a clinician, but self-reported, and the lack of exclusion criteria could have led to a very heterogeneous group of participants. Surprisingly, the sample showed relatively low levels of symptomatology, which could have hindered larger effects of ABCT. In addition, the lack of a control group undermines the strength of our findings. The study outcomes did not include a measure of functional status, such as the FIQ, which is normally considered the primary outcome in studies conducted with fibromyalgia patients.

## 5. Conclusions

This pilot study suggests that ABCT-16 has the potential to be effective in improving anxiety and depressive symptomatology in fibromyalgia patients, and to increase abilities such as self-compassion and decentering. Most improvements lasted 3.5 months after the end of the program, which adds to promising evidence suggesting that ACBT’s effects are maintained at least in the medium term. However, adherence to the intervention was not enhanced compared with previous studies that used the original ABCT protocol. Further studies with larger samples and control groups are necessary to confirm these promising results and to deepen our understanding of other potential psychotherapeutic effects of compassion-based interventions for fibromyalgia patients.

## Figures and Tables

**Table 1 ijerph-19-08152-t001:** Means and standard deviations (SD) of the study outcomes at each assessment point.

	Baseline	Session 8	Posttreatment	Follow-Up
HADS total score HADS Anxiety HADS Depression	18.55 (5.85)	14.00 (4.81)	9.09 (3.24)	10.09 (3.78)
	10.82 (3.92)	8.00 (2.58)	5.91 (2.12)	6.36 (1.69)
	7.73 (2.65)	6.00 (2.94)	3.18 (1.78)	3.73 (2.87)
SCS Self-kindness Common humanity Mindfulness	2.47 (0.56) 2.34 (0.70)	2.99 (0.43) 2.93 (0.66)	3.50 (0.60) 3.57 (0.59)	2.97 (0.57) 3.09 (0.66)
	2.95 (0.67) 2.11 (0.52)	3.23 (0.49) 2.83 (0.53)	3.57 (0.74) 3.36 (0.66)	3.02 (0.85) 2.80 (0.47)
EQ-D	31.91 (5.54)	36.50 (5.74)	40.55 (3.24)	37.73 (3.82)

Note: HADS = Hospital Anxiety and Depression Scale. SCS = Self-Compassion Scale. EQ-D = Experiences Questionnaire.

**Table 2 ijerph-19-08152-t002:** Statistical significance of the changes in the study outcomes.

	From Baseline to Session 8			From Baseline to Posttreatment			From Baseline to Follow-Up		
	Z	*p*	r	Z	*p*	r	Z	*p*	r
HADS total score	−2.43	**0.015**	0.73	−2.94	**0.003**	0.89	−2.81	**0.005**	0.85
HADS Anxiety	−2.39	**0.017**	0.72	−2.95	**0.003**	0.89	−2.94	**0.003**	0.89
HADS Depression	−2.40	**0.016**	0.72	−2.94	**0.003**	0.89	−2.85	**0.004**	0.86
SCS total score	−1.84	0.066	0.55	−2.50	**0.012**	0.75	−1.78	0.075	0.54
Self-kindness	−1.83	0.068	0.55	−2.59	**0.010**	0.78	−2.41	**0.016**	0.73
Common humanity	−1.33	0.182	0.40	−1.69	0.090	0.51	−0.49	0.622	0.15
Mindfulness	−2.40	**0.017**	0.72	−2.71	**0.007**	0.82	−2.32	**0.021**	0.70
EQ-D	−2.30	**0.022**	0.69	−2.67	**0.008**	0.80	−2.49	**0.013**	0.75

Note: in bold, statistically significant results (i.e., *p* < 0.050). HADS = Hospital Anxiety and Depression Scale. SCS = Self-Compassion Scale. EQ-D = Experiences Questionnaire.

## Data Availability

The data that support the findings of this study are available from the corresponding author upon reasonable request.

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
