# Peer review of "Attachment-Based Compassion Therapy for Reducing Anxiety and Depression in Fibromyalgia"

_ijerph, 2022, doi:10.3390/ijerph19138152_

Round 1

Reviewer 1 Report

1) No control group 

2) I am not familiar with this method and the authors should elaborate more about it in order that others will try it . 

3) the participants self reported to have fibromyalgia . No proffessional evaluation of the diagnosis was done which is a graet mistake of the soundness of this pilot . 

The study outcomes did not include a measure of functional status

Author Response

Thank you for your comments.

We acknowledge the lack of a control group as a limitation in line 263.

As regards the methods, the analysis of the effects of the intervention on the different study outcomes is quite generic (we used Wilcoxon’s test owing to the small sample size). The RCI and SCS are also commonly included in similar studies since they give additional information regarding the effect of the intervention. In lines 140–157, we provide a description on how to calculate these indexes, and the reference is presented in line 141 (Jacobson, N.S.; Truax, P. Clinical Significance: A Statistical Approach to Defining Meaningful Change in Psychotherapy Research. J. Consult. Clin. Psychol. 1991, 59, 12–19). These are similar studies that have used the same methodology:

https://pubmed.ncbi.nlm.nih.gov/34868374/

https://pubmed.ncbi.nlm.nih.gov/33246652/

We agree with the reviewer regarding the limitation posed by not having a professional assessment of fibromyalgia in this study. This was underscored as one of the main limitations in lines 260-262.

Likewise, we acknowledge that this study did not include any measure of functional status in lines 264-266.

Reviewer 2 Report

The work is relevant for presenting a therapy for patients with fibromyalgia. The authors suggest that ABCT-16 can be effective for improving anxiety and depressive symptomatology in fibromyalgia patients.

Minor comment:

-  this sense, it is necessary to include in the discussion the possible reasons for the ABCT to present lasting effects after the period of 3 months.

- Check the numbering of the tables in the legend. Table 1 appears twice and the table 2 is absent.

Author Response

Thank you for your kind comments.

Following the reviewer’s suggestion, we have included some new details on why we believe ABCT produces long-term effects in lines 217-221:

The reasoning behind the long-term impact of ABCT could be that this intervention focuses on producing a deep change in the way individuals relate to themselves by promoting the development of a more secure attachment figure. While this therapeutic change might be harder to be produce than others that are more context-related, its effects could last longer once it has been achieved [10].

Thank you for noticing the error in the tables; it has been corrected.

Reviewer 3 Report

This manuscript presents a pilot study aimed at exploring the influence of an adapted version of ABCT on improving anxiety and depression symptoms and enhancing self-compassion and decentering of a sample of patients with fibromyalgia. However, some limitations are noted in this study in terms of selection of participants, protocol application, and results. The limitations are as follows.

1.     Please clarify have you randomly selected participants and what's your response rates? The sample size is too small, and only includes women (11 respondents?), which would likely lead to biased results.

2.     Need to change table 1 title to table 2 titles.

3.     As your results declared most improvements lasted 3.5 months after the end of the program, it might be interesting to add one more assessment point longer than 3.5 months after finishing the program. 

Author Response

Thank you for your comments.

We have included new details on the selection process in the Participants section (line 89-90):

Of the 80 members of the association, 11 women (mean age 54.91, SD = 8.81) volunteered and were recruited after checking their compliance with the following inclusion criteria

Likewise, we acknowledge that the small sample size is the main limitation of this study and that it hinders the generalization of our findings in lines 257-260.

Thank you for noticing the error in the tables; it has been corrected.

We have added as a suggestion for future studies to include new follow-up assessments to verify whether the effects are maintained for periods even longer than 3.5 months (lines 252-256):

Moreover, given that previous studies have already found the effects of ABCT to be maintained after 3 months [16], further studies should consider extending the fol-low-up assessments to at least 12 months in order to verify whether the hypothesized deep changes produced in the attachment figure of the individual remain significant in the longer term.

Reviewer 4 Report

The authors conducted a brief study evaluating the effects of attachment-based compassion therapy (ABCT) in reducing the levels of anxiety in depression in patient with fibromyalgia. This study may be considered for publication after providing appropriate responses to the below mentioned comments.

1. Your introduction section appears somewhat short. I suggest you mention that fibromyalgia does not need to be an only occurring condition, but can also overlap with other pathologies. Suggested reference: https://pubmed.ncbi.nlm.nih.gov/29802812/

2. Methods section: you seem to be missing a table with demographic information for the 11 participants. Just merely providing the number does not give the readers much of an information. Any information about the location and severity of their symptoms? Please elaborate whether any of the study participants have been using any treatment drug(s) for fibromyalgia prior to recruitment. Also, please specify this in the limitations section.

3. Please specify what the participants' compliance was throughout the study. Have any of the patient been excluded, and why?

Author Response

Thank you for your comments.

Following the reviewer’s recommendation, we have included the suggested reference to stress that fibromyalgia often appears concurrently with other conditions (lines 38-40). The introduction is purposefully brief because it was our intention to follow the “Brief Report” guidelines:

It affects approximately 2% of the general population and is often concomitant with other medical disorders, such as gastrointestinal diseases and other pain-related con-ditions, and mental health disorders [2,3,4].

As regards the table with sociodemographic data, it was not included for two reasons: 1) the only data that was asked of the patients was their age and their place of residence (all were from the same region), and 2) we already had presented 2 tables and a supplementary table with an outline of the intervention. Therefore, considering that there was not enough sociodemographic information, and the fact that we already had 2 tables, which is typically the rule for short communications, we believe this table cannot be included.

We fully agree with the reviewer that more details regarding adherence to the intervention were necessary. These have been included in lines 162-166:

3.1. Adherence to the ABCT program

The 11 participants who commenced the intervention were evaluated at every assessment point. The average number of sessions attended was 10.4 (SD = 2.2), and all the participants completed at least half of the program (i.e., attended 8 sessions). On average, the participants completed 64.8% of the sessions.

And a paragraph has been added in the Discussion (lines 241-256):

The ABCT-16 program was an adaptation of an original protocol designed to meet the needs of fibromyalgia patients, who often present with cognitive difficulties (i.e., fibrofog) that hinder their ability to focus on the instructors’ explanations, and who had acknowledged in previous studies that the length of the psychotherapy sessions––around 2 hours––was an obstacle because it was hard for many of them to remain seated for such a long period [27]. Rates of adherence to ABCT-16 in this pilot study were good, given that all patients attended at least 8 of the 16 sessions. The role of the instructors was key in maintaining the patients’ link with the intervention, since they conducted phone calls and sent the session contents and homework via email every time a patient had not been able to attend. Thus, it remains unclear whether adherence to the intervention would have been equally high had the instructors not made such efforts. Future studies using larger samples and control groups could put this to the test. Moreover, given that previous studies have already found the effects of ABCT to be maintained after 3 months [16], further studies should consider extending the fol-low-up assessments to at least 12 months in order to verify whether the hypothesized deep changes produced in the attachment figure of the individual remain significant in the longer term.

Reviewer 5 Report

1. Overall Strengths

The manuscript has not have reach merits. The number of participants is not significance, very short with 11 participants, and poor description of the methodology/statistics. The text is structured and is dificult to follow. there are several language related errors and thus the manuscript requires a thorough checking by a native speaker of English.

2. Importance

The topic is interesting potential to offer more solid evidence based information about the attachment-based compassion therapy for reducing anxiety and depression in fibromyalgia.

Hypotheses are needed. What do the authors expect to find based on previous work? This would also be a good place to justify why this study is needed.

3. Justification/Rationale

The justification of this study could be strengthened by explaining what this study adds to the pre-existing literature in the introduction section and comparing with recently researches of Palomo López  associated to fibromyalgia, quality of life and foot health and women and too relationship of Depression Scores and Ranges in Women Who Suffer From Fibromyalgia by Age Distribution.

Please make it clearer already in the Introduction, what new this study has to offer (i.e. in what way it differs from earlier studies).

4. Methods/Approach

The methodology is not rationale, seriously flawed from the methodological point of view related with the number record type of the design and trial registry before the first participant is enrolled into the study creates a publicly-available record of the researcher’s intentions; including key details such as how many patients they need to recruit, what their outcomes will be, how they intend to measure these outcomes, etc. I appreciate that the authors included a sample size calculation.
Also, Add a study flow chart for the readers ?

5. Results/Findings

The analysis is incomplete in all the tables – what tests were used when data were normally distributed?

The results maybe may be statistically significant, but is it clinically relevant? 

Include p-values in all the tables 

6. Discussion

Discussion is muddled, confusing to follow and repeats somewhat the Introduction. Furthermore, some more shortcomings should be included, particularly the fact the bias found. The authors could also consider some more ideas for further studies.

7. Conclusions

Write this section part again, clearly and include causative conclusions are warranted.

Author Response

Thank you for your comments.

While we are not able to improve the methods of this study, we have tried to improve the quality of the writing and to make our procedures as clear as possible.

We agree with the reviewer about the need to include the study hypotheses, which can now be found in lines 80-85:

The study hypotheses were 1) that ABCT-16 would significantly reduce posttreatment symptomatology and promote “third wave” skills such as self-compassion and decen-tering, 2) that these changes would be maintained by the follow-up assessment (i.e., 3.5 months after the end of the program), and 3) that adherence to the intervention would be similar or higher than that reported by Montero-Marín et al. [16].

The introduction has been rewritten to include new references and to highlight the main difference between the present study and previous ones: using an adaptation of ABCT to improve depressive and anxiety symptoms in a small sample of patients with FM.

The study was designed as a pilot study with the potential to evolve into a randomized controlled trial in the future. However, that did not happen, and therefore there was no register for a wider, sounder study with a larger sample size. It was not possible to calculate sample size since this study did not use a comparator, only the one group.

Wilcoxon’s test was used not only because of the non-normal distribution of the data, but also because of the small sample used. The evident limitations of our study undoubtedly undermine the clinical significance of our findings, which presented moderate effect sizes according to Rosenthal r. We complemented these analyses with the calculation of the reliable change index (RCI) and clinically significant change (SCS). While most patients changed in a reliable way, only 2 achieved a significant change in clinical terms. We acknowledge this in lines 202-207:

However, CSC was more difficult to assess, considering that baseline symptomatology levels were relatively low; only two patients presented high enough HADS “Anxiety” baseline scores, and although they both experienced posttreatment CSC, further re-search using larger samples and with more severe symptomatology is needed to assess with greater accuracy the potential of ABCT-16 to produce clinically significant changes in both anxiety and depressive symptomatology.

P values are represented in table 2, which complements the information in table 1 (raw scores at each time point).

Following the reviewer’s suggestion, we have included new proposals for future studies in lines251-256:

Future studies using larger samples and control groups could put this to the test. Moreover, given that previous studies have already found the effects of ABCT to be maintained after 3 months [16], further studies should consider extending the fol-low-up assessments to at least 12 months in order to verify whether the hypothesized deep changes produced in the attachment figure of the individual remain significant in the longer term.

The conclusions section has been revised (lines 272-277):

However, adherence to the intervention was not enhanced compared to previous studies that used the original ABCT protocol. Further studies with larger samples and control groups are necessary to confirm these promising results and to deepen our un-derstanding of other potential psychotherapeutic effects of compassion-based inter-ventions for fibromyalgia patients.

Round 2

Reviewer 3 Report

Looks good.

Reviewer 4 Report

Thank you for providing answers to my questions. The manuscript can now be accepted in the present form.

Reviewer 5 Report

My opinion about the article remains the same of the first revision of manuscript. Most of the issues that I advanced to you cannot be repaired. A new study would be needed to make these things suitable. The clarifications provided do not solve the problem. Best regards.